# Cytotoxicity of *Callerya speciosa* Fractions against Myeloma and Lymphoma Cell Lines

**DOI:** 10.3390/molecules27072322

**Published:** 2022-04-03

**Authors:** Vu Quang Lam, La Hoang Anh, Nguyen Van Quan, Tran Dang Xuan, Ichiro Hanamura, Kaori Uchino, Sivasundaram Karnan, Akiyoshi Takami

**Affiliations:** 1Division of Hematology, Department of Internal Medicine, Aichi Medical University School of Medicine, Nagakute 480-1195, Japan; quanglamvu1991@gmail.com (V.Q.L.); hanamura@aichi-med-u.ac.jp (I.H.); ksakai@aichi-med-u.ac.jp (K.U.); 2Transdisciplinary Science and Engineering Program, Graduate School of Advanced Science and Engineering, Hiroshima University, Hiroshima 739-8529, Japan; hoanganh6920@gmail.com (L.H.A.); nvquan@hiroshima-u.ac.jp (N.V.Q.); tdxuan@hiroshima-u.ac.jp (T.D.X.); 3Department of Biochemistry, Aichi Medical University, Nagakute 480-1195, Japan; skarnan@aichi-med-u.ac.jp

**Keywords:** *Callerya speciosa*, antioxidants, multiple myeloma, mantle cell lymphoma, liquid chromatography–electrospray ionization–tandem mass spectrometry, cell viability, cell apoptosis

## Abstract

*Callerya speciosa* is widely distributed in tropical and subtropical countries and is traditionally used for preventing numerous disorders. In this study, a bioguided fractionation of ethyl acetate extract (SE) from *C. speciosa* root was carried out to target antioxidant and cytotoxic activities. Of the four fractions (SE1-SE4) obtained by column chromatography, SE4 had the strongest anti-radical ability in the DPPH and ABTS assays (IC_50_ = 0.05 and 0.17 mg/mL, respectively), with results close to butylated hydroxytoluene (BHT), a common antioxidant agent. The cytotoxic activities against the selected cells were analyzed in this study by MTT assay. Accordingly, SE2, SE3, and SE4 significantly inhibited the viability of multiple myeloma cell lines, comprising U266 (IC_50_ = 0.38, 0.09, and 0.11 mg/mL, respectively) and KMS11 (IC_50_ = 0.09, 0.17, and 0.15 mg/mL, respectively), mantle cell lymphoma Mino (IC_50_ = 0.08, 0.16, and 0.15 mg/mL, respectively), and the noncancerous cell line LCL (IC_50_ = 0.40, 0.32, and 0.21 mg/mL, respectively). At a concentration of 125 µg/mL, SE2, SE3, and SE4 induced the cell apoptosis of U266 (32.2%, 53.2%, and 55.6%, respectively), KMS11 (36.9%, 40.8%, and 47.9%, respectively), Mino (36.6%, 39.8%, and 22.0%, respectively), and LCL (12.4%, 17.5%, and 23.5%, respectively) via annexin V assay. The dominant compounds detected in fractions by high-performance liquid chromatography–electrospray ionization–tandem mass spectrometry (HPLC-ESI-MS/MS), were identified as isoflavones. This is the first report describing *C. speciosa* as a promising natural source of antileukemia and antimyeloma agents, which may be useful for the development of blood cancer treatments.

## 1. Introduction

Hematological disorder is a common and dangerous type of cancer with 1.24 million cases every year worldwide [1]. In 2020, the mortality rate of blood cancer was approximately 60% [2]. Among blood cancer cases, multiple myeloma, which accounts for 17% of hematological disorders, is a malignant neoplasm of plasma cells. In this cancer, the structure and function of bone are damaged due to accumulated cells in bone marrow [3]. On the other hand, lymphoma is the most common type of hematological cancer, accounting for 45% of all cases in 2020 [2]. Mantle cell lymphoma is a rare subtype of non-Hodgkin’s lymphoma, with a median survival time of 3 to 5 years [4]. In cancer research, human-derived tumor cell lines have enormously contributed to forming the fundamental knowledge and gaining a deeper understanding of the drug sensitivity. Inhibiting the growth of tumor cells could be a potential method for preventing the progression of cancer. In multiple myeloma, U266 and KMS11 are well-known cell lines, while Mino is typical in mantle cell lymphoma. In addition, noncancerous cells are indispensably used to confirm the cytotoxic selectivity of potential candidates, since they should suppress tumors without harmful effects on normal cells [5]. Thus, LCL, an EBV-transformed lymphoblastoid B-cell line, was selected as a benchmark in this study. This cell line is commonly applied as a standard in various studies of hematological disorders [6]. In addition, previous studies demonstrated a close association between oxidative stress and cancer via inflammatory processes in the body. Particularly, oxidative stress can exacerbate inflammation and vice versa, while inflammation is considered to be the key physiological process in the development of serious chronic diseases, including cancer [7]. Furthermore, cancer treatments that use synthetic drugs have met with complications due to the side effects related to oxidative stress [8]. Therefore, substances with high cytotoxic selectivity against the mentioned tumor cells and potent antioxidant activity could be prospective candidates to develop therapies for multiple myeloma and mantle cell lymphoma.

Among the potential sources for isolating anticancer agents, traditional herbal medicine has received increased attention across a number of disciplines in the last decade. Recent trends in developing various alternative medicines also have led to a proliferation of studies in exploiting benefits from plants. Moreover, plant resources are extremely abundant and well-distributed all over the world [9]. However, to date, only a limited number of herbal medicines have been officially approved for therapeutic purposes. A plant-based product needs to be qualified in all evaluations to demonstrate that its benefits outweigh its risks before the product can be widely accepted as a drug [10]. Thus, instead of lesser-known herbs, common plants, which have a long history of application, are greater candidates for novel alternative therapies. The advantages of this strategy include the use sustainable material resources and the demonstrated safety of plants that have been examined for generations.

Among medicinal plants, *Callerya speciosa* (known as “Sam Nam” in Vietnamese and “Niu Da Li” in Chinese) belongs to the Magnoliopsida class in the Fabaceae family. *C. speciosa* is widely found in the northern area of Vietnam and in several southern provinces in China. In folklore, this plant is known for its ability to cure several diseases such as cough, arthritis, hepatitis, irregular menstruation, numbness of the wrists or knees, low blood flow rates, anemia, tuberculosis, chronic bronchitis, and chronic hepatitis [11,12,13,14,15]. Additionally, *C. speciosa* root is a cooking ingredient that plays an important role in several local dishes and is believed to be highly nutritious. On the other hand, very few published projects have evaluated the effects of *C. speciosa* root’s bioactivities both in vitro and in vivo. Moreover, the *Callerya* genus is well acknowledged for its pharmaceutical properties due to the existence of numerous valuable secondary compounds such as alkaloids, triterpenoids, coumarins, flavonoids, isoflavonoids, phenols, and phytosterols [16]. For instance, millepurone from *Callerya atropurpurea* and osajin from *Callerya auriculata* displayed antioxidant and antitumor properties [16]. In particular, *C. speciosa* was reported as rich in saponins, flavonoids, phenolic glycosides, polysaccharides, and formononetin, which revealed numerous biological benefits for humans [13,17]. However, there has been no research on phytochemical profiles related to cytotoxic potentials of *C. speciosa* against multiple types of hematological cancer.

The aforementioned rationales prompted us to evaluate the antioxidant and cytotoxic activities of *C. speciosa* fractions against LCL, U266, KMS11, and Mino. Moreover, the phytochemical profiles of this valuable plant were identified using liquid chromatography–electrospray ionization–tandem mass spectrometry (LC-ESI-MS/MS). These findings are expected to make an important contribution to the development of foods and alternative medicines in the future.

## 2. Results

### 2.1. Yields and Total Phenolic Contents (TPC) and Antioxidant Activities of Extracts from C. speciosa Roots

From a total of 1.284 kg of *C. speciosa* dried roots, 223.87 g (17.44%) of methanolic extract (SM) was obtained. After fractionation by different solvents, the yields of hexane (SH), ethyl acetate (SE), butanol (SB), and water (SW) extracts were 5.4 (0.42%), 4.84 (0.38%), 13.6 (1.06%), and 114.11 g (8.89%), respectively. The TPC and antioxidant activities (DPPH and ABTS assays) are presented in Table 1.

The highest TPC (1.77 mg GAE/g DW) was found in the SM extract. The SW sample achieved second place with a TPC of 0.84 mg GAE/g DW, followed by the SB, SE, and SH fractional extracts, with TPCs of 0.18, 0.09, and 0.06 mg GAE/g DW, respectively.

In the DPPH assay, the strongest extract was SE, with an IC_50_ value of 0.20 mg/mL The IC_50_ values of the remaining extracts including SM, SH, SB, and SW were 0.79, 0.82, 2.43, and 15.20 mg/mL, respectively. The positive control BHT demonstrated significant scavenging activity against DPPH with an IC_50_ value of 0.02 mg/mL.

In the ABTS assay, the IC_50_ values of the extracts ranged from 0.39 (SE) to 5.41 mg/mL (SW). The extracts comprising SM, SH, and SB had IC_50_ values of 1.75, 1.84, and 3.98 mg/mL, respectively. The IC_50_ value of BHT was 0.08 mg/mL.

Based on the results from the in vitro antiradical assays, the SE extract presented the strongest inhibitory capacities. Therefore, this sample was selected for further fractionation of the bioactive components involved in antioxidant and cytotoxic activities.

### 2.2. Fractionation by Column Chromatography

The SE extract (3.29 g) was fractionated by column chromatography over silica gel. Based on the TLC results, similar fractions including F1–F20, F21–F37, F38–F59, and F60–F139 were combined to retrieve a total of 4 samples encoded as SE1, SE2, SE3, and SE4, respectively (Table 2). The highest yield was found in SE4 (0.14 g, 4.25%), followed by SE2, SE3, and SE1 with yields of 0.07 (2.12%), 0.04 (1.21%), and 0.01 g (0.30%), respectively.

### 2.3. Antioxidant Capacities of Fractions from SE Extracts

According to the results of the DPPH radical-scavenging assay (Figure 1), SE4 had the most potent scavenging ability against DPPH radicals, with an IC_50_ value of 0.05 mg/mL, which was close to the positive control, BHT (IC_50_ = 0.02 mg/mL). The second-strongest fraction was SE3 (IC_50_ = 0.09 mg/mL), followed by SE2 (IC_50_ = 0.21 mg/mL). SE1 was the weakest sample inhibiting DPPH radicals, with an IC_50_ of 1.30 mg/mL.

For ABTS cation-scavenging outcomes (Figure 1), SE4 presented the strongest scavenging activity among the fractions against ABTS cations (IC_50_ = 0.17 mg/mL), while the IC_50_ values of SE3, SE2, and SE1 were 0.26, 0.37, and 1.55 mg/mL, respectively.

In general, the SE4 fraction presented the strongest antioxidant activity in both the DPPH and ABTS assays.

### 2.4. Cytotoxic Activities of Fractions from SE Extracts

The in vitro cytotoxic activities against the tumor cell lines, including multiple myeloma (U266 and KMS11) and B-cell lymphoma (Mino) in fractions from the SE extract, are expressed as IC_50_ values in Table 3. The lymphoblastoid cell line LCL was tested as the noncancerous control. Moreover, doxorubicin was applied as a standard inhibitor with IC_50_ values of 5.56, 0.13, 0.46, and 3.09 µg/mL for the LCL, U266, KMS11, and Mino cell lines, respectively.

For LCL, SE1 exhibited an insignificant effect. At a concentration of 500 µg/mL, the inhibition percentage of SE1 against LCL was 23.20%. The SE2, SE3, and SE4 samples began inhibiting the growth of LCL cells at high concentrations (IC_50_ = 398.58, 324.40, and 205.42 µg/mL, respectively).

In the case of U266, the expansion of U266 cells was prevented by all the tested samples, in which the IC_50_ values ranged from 87.86 to 383.25 µg/mL. Notably, SE3 exhibited the most potent cytotoxicity against U266 cells (IC_50_ = 87.83 µg/mL), followed by SE4 (IC_50_ = 112.93 µg/mL). The fractions including SE1 and SE2 showed suppression at an average level (IC_50_ = 353.55 and 383.25 µg/mL, respectively). In addition, SE1 revealed trivial effects on normal cells, while SE3 displayed the highest selectivity (SI = 3.69) for the cytotoxicity against U266 cells, followed by SE4 and SE2 (SI = 1.82 and 1.04, respectively). Doxorubicin was highly selective for preventing U266 cells, with an SI of 42.77.

The cytotoxic effects of the tested fractions on KMS11 indicated that all the fractions significantly inhibited cell growth. Among them, the strongest fraction was SE2 (IC_50_ = 94.93 µg/mL). The remaining samples including SE1, SE3, and SE4 presented insignificant differences, with IC_50_ values of 179.04, 169.06, and 152.91 µg/mL, respectively. Among the tested fractions, SE2 was the most selective fraction in inhibiting KMS11 cells, with an SI of 4.20. The SI values of SE3 and SE4 were 1.92 and 1.34, respectively. Doxorubicin had an SI of 12.09.

For the Mino inhibitory assay, SE1 showed slight inhibition (IC_50_ = 335.54 µg/mL). Conversely, the other fractions showed strong suppression of Mino cell viability. Notably, SE2 exhibited the highest suppression (IC_50_ = 76.65 µg/mL). The following samples including SE3 and SE4 had IC_50_ values of 159.82 and 151.69 µg/mL, respectively. Remarkably, SE2 and SE3 (SI = 5.20 and 2.03, respectively) achieved higher selectivity indexes than the drug doxorubicin (SI = 1.80). The SI of SE 4 was 1.35.

The dose–response curves of all fractions against blood cancer cells are presented in Figure 2.

As shown in Figure 2, the fraction SE1 inhibited KMS11 from a concentration of 62.5 μg/mL. However, until reaching a concentration of 500 μg/mL, mild cytotoxicity to LCL was recorded. Similarly, SE2 showed highly potent cytotoxic activities against Mino and KMS11 (inhibition of up to 50%) at a low concentration of 62.5 μg/mL. SE2 slightly inhibited the development of LCL from 62.5 to 250 μg/mL. At a concentration of 500 μg/mL, a strong suppression of LCL was observed. Interestingly, from a concentration of 250 μg/mL onward, the cytotoxic activity of SE2 insignificantly increased, while toxicity to LCL began to be observed. Therefore, the appropriate concentration for SE2 was under 250 μg/mL. Moreover, SE3 substantially suppressed tumor cell lines, especially U266 at 62.5 μg/mL. However, SE3 was reported to be more toxic to LCL (inhibition of up to 50%) than SE1 and SE2 at 250 μg/mL. SE4 also inhibited tumor cell lines strongly from 62.5 to 125 μg/mL. However, this fraction was observed to exert significant suppression from 125 to 250 μg/mL. Notably, SE3 and SE4 prevented tumor cell lines substantially in a concentration range of 62.5–125 μg/mL. However, from 125 μg/mL onward, the inhibitory effect increased slightly and began producing LCL cytotoxicity.

Overall, most fractions exhibited potent prevention in the tested blood cancer cells, except for SE1. In addition, SE1 showed weak antioxidant abilities, as presented in Figure 1. Therefore, SE2, SE3, and SE4 were selected for a further apoptosis assay. Based on the dose–response curves of the fractions against blood cancer cells, the tested concentration for the apoptosis assay was 125 µg/mL.

### 2.5. Apoptosis-Inducing Activities of Fractions from SE Extracts

From the results of cytotoxic activity, the SE2, SE3, and SE4 fractions (125 µg/mL) were selected to evaluate the ability to induce apoptosis in Mino, U266, and KMS11, and LCL was used as a control. Twenty-four hours after adding the fractions into the culture medium, the result was examined with a flow cytometer (Figure 3).

All fractions showed the capacity to induce cell apoptosis against the three cell lines after 24 h (Figure 3). LCL cells were least affected by the SE2, SE3, and SE4 fractions, with the percentage of cell apoptosis increasing by 12.4%, 17.5%, and 23.5%, respectively.

Closer inspection of the tests with SE2 demonstrated a dramatic effect of inducing apoptosis against Mino and U266, with 36.6% and 32.0%, respectively. In the case of SE3, Mino cells exhibited a similar reaction to SE2, while U266 cells were surprisingly sensitive, with this fraction increasing to 53.0%.

Interestingly, the SE4 fraction presented a very poor impact against Mino cells, as only 22.0% of apoptosis cells were recognized—even lower than the proportion of LCL cells. U266 cells maintained a strong reaction with SE4, while over half of the cells experienced the process of apoptosis.

### 2.6. Identification of Phytochemical Components of Active Fractions from SE Extract by LC-ESI-MS/MS

The results from LC-ESI-MS/MS indicated that a total of 26 tentative compounds were detected in the tested fractions belonging to various groups. The most abundant group was isoflavones with six detected compounds, followed by triterpenes with three identified compounds. Other groups of compounds were keto acids, lactones, ethers, resorcinols, indoles, lignans, esters, flavanones, flavones, pterocarpans, amines, fatty acid esters, fatty acyls, triterpenoids, carboximidic acids, and ergostane steroids. Detailed information is provided in Table 4.

As shown in Table 4, six compounds were identified in SE1, including keto acids, isoflavones, flavones, pterocarpans, fatty acid esters, and fatty acyls. In SE2, nine compounds were detected. Among them, lignans and isoflavones were the most dominant groups, with two detected compounds. Other groups included indoles, amines, and fatty acyls. The results of SE3 showed that this fraction was rich in isoflavones, with four identified compounds. The second most abundant group of compounds in SE3 was triterpenes, with two detected compounds. Other compounds of SE3 were classified into the groups of lactones, resorcinols, and fatty acyls. In the case of SE4, we detected a total of 12 compounds belonging to 10 groups, in which isoflavones and triterpenes were the most abundant compounds. Other detected groups included ethers, esters, flavanones, amines, fatty acyls, triterpenoids, carboximidic acids, and ergostane steroids.

## 3. Discussion

Recently, plant-based products have been widely deployed in many countries around the world for therapeutic purposes. However, to date, there has been no research on the cytotoxic properties of *C. speciosa* against hematological cancer. In previous studies, numerous plant species presented potential anticancer activities against blood cancer. For example, MTT assays showed that ethanolic extracts from *Origanum syriacum* L. and *Thymus vulgaris* L. inhibited the viability of the human leukemia cell THP-1 (IC_50_ = 2.126 and 0.1569 mg/mL, respectively) [18]. In the present research, the expansion of mantle cell lymphoma (Mino) and myeloma (U266 and KMS11) cell lines was significantly reduced by fractional extracts from *C. speciosa* roots, which presented lower IC_50_ values ranging from 0.08 to 0.38 mg/mL (Table 3). According to Indrayanto et al. [5], an ideal candidate for developing an anticancer drug should have a high cytotoxic selectivity of strong inhibition of tumors and possess very low toxicity against normal cells, in which substances with selectivity indexes (SI) greater than or equal to 3 can be classified as promising candidates for further in vivo and clinical tests [19]. The present study revealed that SE2 and SE3 had a high cytotoxic selectivity against the tested cancer cell lines (SI ranged from 3.69 to 5.20). Significantly, the SI of SE2 against Mino was approximately 3-fold higher than that of doxorubicin (Table 3). This finding might be fundamental for further anticancer research.

Through the annexin V assay, we determined for the first time that obtained fractions from *C. speciosa* roots may suppress blood cancer cells by promoting cell apoptosis (percentages of apoptosis in cancer cells ranged from 22.0% to 55.6%). Meanwhile, these fractions had trivial effects on the noncancerous cell line LCL (the percentages of apoptotic cells ranged from 12.4% to 23.5%) (Figure 3). Notably, apoptosis is an important mechanism for natural cell death [20]. The loss of apoptosis occurs in cancer, which leads cancer cells to escape from death, thereby giving cancer cells more opportunities to grow and invade [20]. Chain events of apoptosis can be prevented by increasing the expression of antiapoptotic factors and/or decreasing the expression of proapoptotic factors, which can yield internal resistance to the most common chemotherapy, whereas substances inducing apoptotic signals can eliminate cancer cells. Accordingly, apoptosis is considered a major target in the development of effective anticancer therapy and represents the most successful nonsurgical treatment for all cancers to manage or stop the uncontrolled growth of cancer cells [20]. Therefore, the fractions obtained in the current study may be a potential source for the development of cancer treatments, since they could eliminate tumors by activating the apoptotic process.

In addition to cytotoxic properties, the antioxidant activities of the achieved fractions were another essential component of this research. Remarkably, oxidative stress was proven to be correlated closely with cancer [21], and numerous reports have discussed this correlation. Accordingly, oxidative stress can exacerbate the inflammatory process, which mediates most chronic diseases, including cancer. The existence and contribution of oxidative stress in cancer were clarified in previous studies [22]. Most notably, cancer treatments are complicated by the toxic side effects of drugs associated with oxidative stress. The toxicity of doxorubicin is a typical instance of this issue, and relates to the oxidative stress caused by increasing free radicals [8]. Thus, substances with synergistic antioxidant and cytotoxic effects may be excellent candidates for developing efficient cancer therapies. Basically, the antioxidant abilities of plant extracts can be determined by phenolic content [23], which can readily release hydrogen donors to naturalize free radicals. However, TPC might not be the only factor that regulated the tested fractions’ antioxidant capacities in this study. Other contributors, such as functional groups of individual compounds, might be also important. In particular, the number of free hydrogen donors might determine the antioxidant activity [24,25]. Based on the results shown in Table 1, SM contained the highest TPC among tested extracts; however, the SE extract exhibited the strongest antioxidant property. This implied that the liquid–liquid phase fraction might separate and congregate phytochemicals with greater hydrogen donors into the ethyl acetate extract (SE) while such active groups were conjugated with others such as fatty acids or nonactive groups in the methanolic extract (SM), which might have caused a reduction in the antioxidant activity of SM.

In the present research, the chemical profiles of the most active fractions from the SE extract of *C. speciosa* roots were analyzed by LC-ESI-MS/MS (Table 4). Among the identified compounds, flavonoids were found to be the major group, which might have played an important role in antioxidant activity of the tested samples [26]. Remarkably, fractions that contained more flavonoid compounds (SE2, SE3, SE4) exerted stronger antioxidant capacities than ones that comprised lesser flavonoids (SE1) (see Figure 1 and Table 4). In the other considerations, numerous isoflavones were verified for antioxidant activity [27,28]. Calycosin, which displayed an antioxidant potential by eliminating free radicals [27], was found in SE4. Another isoflavone; namely, formononetin detected in SE3, was acknowledged as a potent antioxidant agent [28]. Therefore, isoflavones may have played a major role in the antiradical abilities of the obtained fractions. Additionally, the flavanone liquiritigenin was detected only in SE4, the strongest antioxidant fraction (Figure 1 and Table 4). Liquiritigenin may have been the main contributor to the potent radical scavenging capacity of SE4, which would be in line with the previous research [29]. In fact, the interactions between compounds may have been more important than individual compound in determining biological activity [24,30,31]. In the current study, triterpenes and lignans were also detected as abundant compounds (Table 4) that have been widely reported to have various biological activities, including antioxidant activities [32]. Thus, interactions between flavonoids and triterpenes may have played an important role in radical scavenging for the most potent fractions of SE3 and SE4 (Figure 1), while the weaker antioxidant activity of SE2 (Figure 1) may have been determined by the gathering of flavonoids and lignans. Nevertheless, the interaction mechanisms between compounds resulting in their antioxidant capacity need deeper investigations.

In addition to antioxidant activities, some detected compounds also manifested cytotoxic properties against various cancer cell lines. These compounds included calycosin [33], liquiritigenin [29], and ergosterol peroxide [34,35] in SE4; pinoresinol [36,37] in SE2; and maackiain [38] and medicarpin [39] in SE1 and SE2, respectively. Particularly, calycosin may relate to modulation of the MAPK signaling pathway, which regulates both cell proliferation and apoptosis [33], while maackiain showed preventative effects on human promyelotic leukemia HL-60 cells by motivating cell apoptosis [38]. Medicarpin is involved in increasing myeloid leukemia cell apoptosis [39]. These flavonoids may have been the major determinants of cytotoxic effects among the tested fractions in this study via the cell apoptosis pathway. Notably, previous studies reported that phytoflavonoids displayed no harmful effects on normal myeloid, peripheral blood, and epithelial cell lines [40]. Therefore, the potent suppression of the tumor cell lines U266, KMS11, and Mino—and the slight effects on the noncancerous cell line LCL of the tested fractions (Table 3, Figure 2 and Figure 3)—may have been due to the dominant presence of flavonoids in the tested fractions. In addition to the important contributions of flavonoids, the various effects of different fractions could be explained by the interactions between the fractions and other detected groups of triterpenes and lignans in the mixtures. Notably, the triterpene betulin exerted cytotoxic effects on the gastric cancer cell line (SGC-7901) by causing apoptosis and G0/G1 arrest of the cell cycle, and might be traced through ROS-mediated mitochondrial dysfunction pathways [41]. Moreover, pinoresinol belonging to lignans exerted anti-breast-cancer activity with different estrogen-receptor statuses [36]. There are many factors involved in apoptotic mechanisms, such as ligands for death receptors, BCL-2 inhibitors, XIAP inhibitors, and alkylphospholipid-like molecules [20]. Hence, further research should be conducted to clarify the main targets of isolated components from *C. speciosa* roots in preventing cancer cells by promoting apoptosis. Along with apoptotic process, cell-cycle arrest may be activated by fractions isolated from *C. speciosa* roots.

In general, the constituents isolated from *C. speciosa* showed high potential in their antioxidant and cytotoxic properties under in vitro approaches. Moreover, the concomitant appearance of isoflavones, triterpenes, and lignans may have contributed to the biological activities. However, further analytical methods should be carried out to comprehensively identify the phytochemical components of the species. For example, flavonoids, phenolics, phytosterols, saponins, alkaloids, polysaccharides, terpenoids, and resins were reported as the main phytochemicals that linked to therapeutic potentials of various *Callerya* species [16]. In addition, future studies focusing on isolation and purification of these potential bioactive compounds from *C. speciosa* should be performed to confirm their medicinal and pharmaceutical properties. On the other hand, the variations in the biological activities and amount of candidate compounds in the digestive system should be considered to ensure their bioaccessibility and bioavailability in the body [42]. Another concern in generating natural products for therapeutic uses is that such products must satisfy the requirements of benefit over risk [10]. In particular, the effective concentration of targeted products should present the strongest cytotoxic activities without cytotoxicity for healthy cells. Moreover, the hepatotoxic effects of using products derived from medicinal plants on humans should be elaborated in the concern of proper evaluation of phytotoxin-induced liver injury [43]. Thus, in vivo and clinical tests are required in future research on generating novel natural drugs from *C. speciosa* roots. The present findings are expected to contribute to the development of blood cancer therapy.

## 4. Materials and Methods

### 4.1. Materials

#### 4.1.1. Plant Materials

*C. speciosa* roots were kindly provided by Dong Thi Kim Cuc (Agricultural Genetics Institute, Hanoi, Vietnam). In 2019, *C. speciosa* roots were collected from Danh Mountain, Bac Giang Province, Vietnam. The identification of the plant species was based on the morphological characteristics published by the Vietnam Plant Data Center (http://www.botanyvn.com, accessed on 17 January 2019) and The Plant List (http://www.theplantlist.org, accessed on 17 January 2019) as major references, and was authenticated by the Agricultural Genetics Institute. The specimen (Casp_AGI2019) was deposited at the same institute.

#### 4.1.2. Chemicals and Cell Lines

Extraction solvents, including methanol, hexane, ethyl acetate, butanol, and chloroform (Junsei Chemical Co., Ltd., Tokyo, Japan), were used to achieve the bioactive fractions in this study. Gallic acid, rutin, and butylated hydroxytoluene (BHT) (Kanto Chemical Co., Inc., Tokyo, Japan) were applied as standards in the experiments. Chemicals including sodium hypochlorite (NaClO), Folin–Ciocalteu’s reagent, sodium carbonate (Na_2_CO_3_), aluminum chloride (AlCl_3_), 2,2′-azinobis-(3-ethylbenzothiazoline-6-sulfonic acid) (ABTS), potassium persulfate (K_2_S_2_O_8_), 2,2-diphenyl-1-picrylhydrazyl (DPPH), sodium acetate (CH_3_COONa), dimethyl sulfoxide (DMSO), hydrochloric acid (HCl) (Kanto Chemical Co., Inc., Tokyo, Japan), 3-(4,5-dimethylthiazolyl)2,5-diphenyl-tetrazolium bromide (MTT), Iscove’s Modified Dulbecco’s Medium (IMDM), and cell lysis buffer (Sigma-Aldrich, St. Louis, MO, USA) were used for the biological tests in the present study.

The following cell lines were purchased from ATCC (Manassas, VA, USA): U266 (number: TIB-196™) and Mino (number: CRL-3000™). KMS-11 was obtained from the Japanese Collection of Research Bioresource Cell Bank (number: JCRB1179). The human B-cell-derived LCL cell line was established from monocytes of healthy donors, and kindly donated by Sonta Shinichi (Division of Cytogenetics, Department of Genetics, Institute for Developmental Research, Aichi Human Service Center, Kasugai, Aichi, Japan) [6].

### 4.2. Sample Preparation and Extraction

The *C. speciosa* roots were preliminarily washed with sterilized water before soaking with sodium hypochlorite (NaClO) 0.5% to eliminate impurities. Then, the sample was rewashed several times using distilled water to remove sodium hypochlorite. Afterward, the clean sample was dried for a week at 40 °C using an oven, and 1.2 kg of finely ground dried root sample powder was extracted in triplicate with 3 L methanol. After that, the methanolic extracts were combined and filtered through filter papers. Subsequently, the achieved solution was subjected to a vacuum evaporator at 50 °C to remove the solvent and obtain the methanol crude extract (SM, 223.87 g). The SM extract was then fractionated to retrieve hexane (SH, 5.4 g), ethyl acetate (SE, 4.84 g), butanol (SB, 13.6 g), and water (SW, 13.6 g) extracts via liquid–liquid phase extractions. The achieved fractional extracts were preliminarily tested for biological activities. Accordingly, the strongest active fraction (SE) was loaded into a column over silica gel (70–230 mesh) to separate the bioactive components (SE1–SE4, see Table 2). The mobile solvent was composed of chloroform and methanol (*v/v*). The decrease in chloroform was in line with the increase in methanol. The fraction separation was based on thin-layer chromatography (TLC) tests, as described previously [44].

### 4.3. Determination of Total Phenolic Content

The total phenolic content (TPC) of the obtained samples from *C. speciosa* was determined by the Folin–Ciocalteu method with some modifications, as presented in our previous study [30]. Initially, 20 µL of the sample and 100 µL of 10% Folin–Ciocalteu’s reagent were blended, immediately followed by the addition of 80 µL Na_2_CO_3_ 7.5% (*w/v*). This combination was then incubated in the dark at room temperature for 30 min. The calculation of TPC was based on the recorded absorbance at 765 nm and expressed as the milligram of gallic acid equivalent per one gram of sample dry weight (mg GAE/g DW).

### 4.4. In Vitro Antioxidant Capacity Assays

The antioxidant abilities of the extracts and achieved fractions from *C. speciosa* were determined by measuring the antiradical capacity via DPPH and ABTS assays, as described in a previous study [24]. The inhibition of DPPH and ABTS radicals was determined based on the discoloration of the final solution with the *C. speciosa* fraction treatments and evaluated as the decreased percentage of absorbance in a comparison with the negative control (MeOH) at 517 nm and 734 nm, respectively. Butylated hydroxytoluene (BHT) was used as a standard inhibitor. A dose–response curve (linear equation) for radical-scavenging activity was established by applying different concentrations of the sample or positive control. Based on the curve, the concentration inhibiting 50% of radicals was calculated and expressed as the IC_50_ value.

### 4.5. MTT Assays

The cell growth rate was determined based on the cell metabolic activity according to an MTT assay. The cells (5 × 10^3^ cells per well) were seeded with 90 µL of culture media into a 96-well plate and cultured (37 °C, 5% CO_2_) for 24 h. The media used for culturing was IMDM (Sigma-Aldrich) containing 10% fetal bovine serum, 5 mM L-glutamine, 100 IU/mL penicillin, and 100 µg/mL streptomycin. Next, samples (10 µL) with different dilution concentrations were added to each well. After 48 h, 10 µL of the MTT solution (5 mg/mL, Sigma-Aldrich) was added to each well, and the cells were incubated at 37 °C for 4 h. Then, 100 µL of cell lysis buffer (10% SDS in 0.01 M HCl) was added to each well to dissolve the colored formazan crystals produced by the MTT assay. The negative control was produced by adding 10 microliters of the culture media instead of the sample, while doxorubicin, a well-known anticancer medicine, was used as the positive control. A SpectraMAX M5 spectrophotometer (Molecular Devices, Sunnyvale, CA, USA) was applied to measure the absorbance at 595 nm. The formular for calculating cytotoxic effects of samples/standards on the tested cell lines was as follows:(1)$$\mathrm{Inhibitory}\;\mathrm{ability}\;(\%)=\frac{C-S}{C}\;\times \;100$$
where C is the absorbance of negative control solution, and S is the absorbance of tested sample/or positive control solution.

The dose–response curves and IC_50_ values of the samples against the tested cell lines were established to compare samples using the same method in the antiradical assays. The selectivity index (SI) of fractions and standard inhibitor was calculated based on the formula below:(2)$$\mathrm{Selectivity}\;\mathrm{index}=\frac{A}{B}$$
where A is the IC_50_ value against noncancerous cells, and B is the IC_50_ value against cancer cells.

### 4.6. Annexin V Assay

The cells were cultured in 6-well plates (5 × 10^5^ cells/well) for 24 h, followed by incubation with fluorescein isothiocyanate (FITC)-conjugated annexin V (Biolegend, San Diego, CA, USA) and propidium iodide (PI) at room temperature for 15 min. Fluorescence intensities of FITC and PI were determined via flow cytometric analysis on a FACSCanto II instrument (BD, Franklin Lakes, NJ, USA).

### 4.7. Identification of Phytochemical Components by Liquid Chromatography–Electrospray Ionization–Tandem Mass Spectrometry (LC-ESI-MS/MS)

The bioactive components of the SE fractions from *C. speciosa* roots were identified by injecting 3 μL of a sample into an Acquity UPLC^®^ BEH C18 (1.7 μm, 50 × 2.1 mm i.d.) column (Waters Cooperation, Milford, MA, USA). The analytical system involved a connection between an LTQ Orbitrap XL mass spectrometer supplied by Thermo Fisher Scientific, Waltham, MA, United States, and a source of electrospray ionization (ESI) (Thermo Fisher Scientific, Waltham, MA, United States). The ESI conditions were set as follows: ion spray voltage (4.5 kV), sheath gas flow rate (55 arb), and aux gas flow rate (15 arb). The mobile phase consisted of solution A (0.1% aqueous formic acid) and solution B (100% acetonitrile), which were adjusted in a gradient program as follows: 5% B for 0–2 min, a linear increase from 5 to 70% B for 2–8 min, 100% B for 8–9 min (maintained for 0.5 min), 100% B to 5% for 9.5–10 min, and equilibration for 2 min. The measurement time was 10 min with a flow rate of 0.3 mL/min under room-temperature conditions. The mass spectra were recorded at a resolution of 60,000 with a scan range of 100–1000 mass-to-charge (m/z). The full-scan and data-dependent scan spectra for MS/MS on the positive model were acquired and processed using the Xcalibur 2.2 SP1 software integrated with the NIST 20 database. The online database Pubchem (National Center for Biotechnology Information, U.S. National Library of Medicine, Bethesda MD, USA) and the literature were used as references for the MS/MS spectra.

### 4.8. Statistical Analysis

All data were built using the Minitab 16.0 software (Minitab Inc., State College, PA, USA). All tests were performed in triplicate, and the results are displayed as the means ± standard deviation (SD). Significant differences between the tests were determined by a one-way ANOVA using Tukey’s test with *p* < 0.05.

## 5. Conclusions

In this study, the bioguided fractionation of ethyl acetate extract from *C. speciosa* efficiently yielded prospective portions with potent antioxidant and cytotoxic properties. Among the obtained fractions, SE4 showed the most notable antioxidant activity, followed by SE3. Moreover, all fractions exhibited potential cytotoxic selectivity against the myeloma and lymphoma cell lines. Remarkably, SE2, SE3, and SE4 demonstrated significant apoptosis activity on the tested cancer cell lines (U266, KMS11, and Mino). SE2, SE3, and SE4 and their major compounds, including isoflavones, triterpenes, and lignans, may represent prospective sources to produce natural anticancer drugs for multiple types of myeloma and mantle cell lymphoma. Accordingly, isolation and purification of the major phytochemicals responsible for the tested bioactivities should be conducted, followed by in vivo and clinical tests to confirm their efficiency and bioavailability in the development of blood cancer therapeutics.

## Figures and Tables

**Figure 1 molecules-27-02322-f001:**
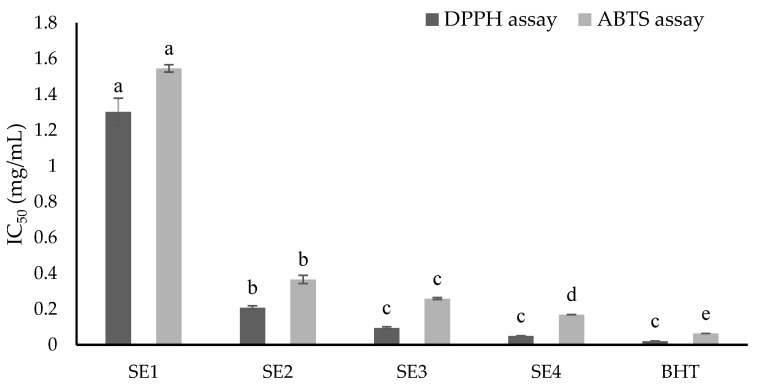
Antiradical activities of obtained fractions from SE extracts. Values enclosed with columns (same colors) followed by similar letters (a, b, c, d, e) indicate insignificant differences at *p* < 0.05.

**Figure 2 molecules-27-02322-f002:**
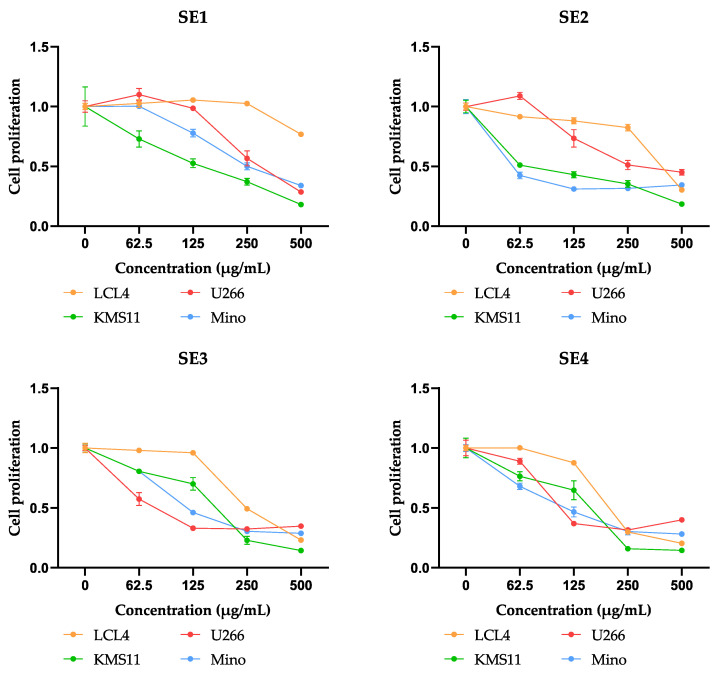
Dose–response curves of obtained fractions (SE1-SE4) from the ethyl acetate (SE) extract of *C. speciosa* for cytotoxic activities against LCL, U266, KMS11, and Mino cell lines.

**Figure 3 molecules-27-02322-f003:**
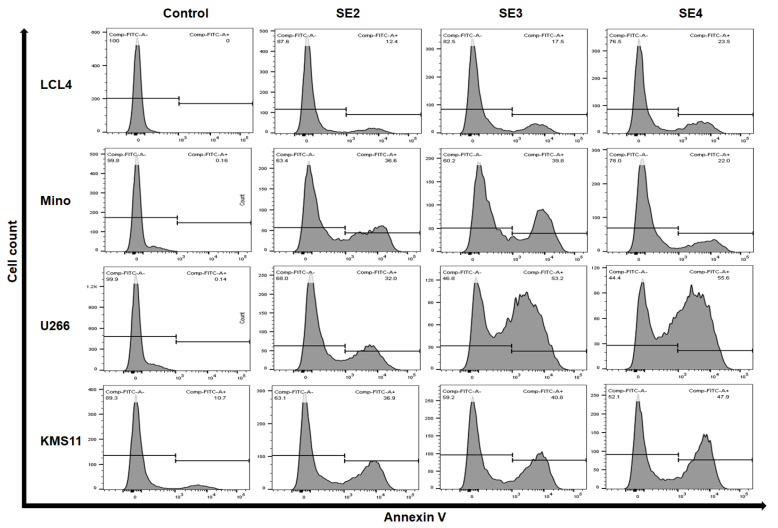
Apoptosis-inducing effect of SE2, SE3, and SE4 fractions against LCL, Mino, and U266 cell lines after 24 h. Cell count shows the number of cells, Comp_FITC_A - represents non-apoptotic cells, while Comp_FITC_A+ indicates apoptotic population.

**Table 1 molecules-27-02322-t001:** Total phenolic contents (TPC) and antioxidant activity of extracts from *C. speciosa*.

Sample	TPC (mg GAE/g DW)	DPPH Assay IC_50_ (mg/mL)	ABTS Assay IC_50_ (mg/mL)
SM	1.77 ± 0.29 a	0.79 ± 0.06 c	1.75 ± 0.14 c
SH	0.06 ± 0.01 c	0.82 ± 0.04 c	1.84 ± 0.05 c
SE	0.09 ± 0.02 c	0.20 ± 0.01 c	0.39 ± 0.01 d
SB	0.18 ± 0.01 b	2.43 ± 0.08 b	3.98 ± 0.06 b
SW	0.84 ± 0.05 b	15.20 ± 0.76 a	5.41 ± 0.25 a
BHT	-	0.02 ± 0.00 d	0.08 ± 0.00 e

Data are presented as the means ± standard deviation (SD). Means within a column followed by different superscript letters (^a, b, c, d, e^) were significantly different at *p* < 0.05. TPC, total phenolic content; GAE, gallic acid equivalent; DW, dry weight, SM, total methanolic extract; SH, hexane extract, SE, ethyl acetate extract; SB, butanol extract; SW, water extract; BHT, butylated hydroxytoluene; -, not determined.

**Table 2 molecules-27-02322-t002:** Column chromatography process of SE extract and obtained fractions’ yields.

No.	Fractions	Mobile Solvent	Code	Amount (g)	Yield (%)
1	F1-F20	C 100%; CM 0.1%	SE1	0.01	0.30
2	F21-F37	C 100%; CM 0.1%	SE2	0.07	2.12
3	F38-F59	CM 0.2%	SE3	0.04	1.21
4	F60-F139	CM 0.2%; CM 0.5%; CM 1%	SE4	0.14	4.25

C, chloroform; CM, chloroform:methanol (*v/v*); M, methanol.

**Table 3 molecules-27-02322-t003:** Cytotoxic activities of fractions from SE extracts.

Sample	LCL	U266		KMS11		Mino	
	IC_50_ (µg/mL)	IC_50_ (µg/mL)	SI	IC_50_ (µg/mL)	SI	IC_50_ (µg/mL)	SI
SE1	ne	353.55 ± 9.82 ^b^	-	179.04 ± 35.26 ^a^	-	335.54 ± 11.12 ^a^	-
SE2	398.58 ± 9.06 ^a^	383.25 ± 6.65 ^a^	1.04	94.93 ± 3.78 ^b^	4.20	76.65 ± 0.84 ^c^	5.20
SE3	324.40 ± 11.67 ^b^	87.86 ± 4.52 ^d^	3.69	169.06 ± 11.54 ^a^	1.92	159.82 ± 5.24 ^b^	2.03
SE4	205.42 ± 6.52 ^c^	112.93 ± 5.06 ^c^	1.82	152.91 ± 6.28 ^a^	1.34	151.69 ± 0.83 ^b^	1.35
Doxorubicin	5.56 ± 0.75 ^d^	0.13 ± 0.00 ^e^	42.77	0.46 ± 0.01 ^c^	12.09	3.09 ± 0.12 ^d^	1.80

Data presented as means ± standard deviation (SD). Means within a column followed by different superscript letters (^a, b, c, d, e^) were significantly different at *p* < 0.05. LCL, non-cancerous B-cell derived cell line; U266 and KMS11, typical cell lines for multiple myeloma; Mino, typical cell line for mantle cell lymphoma; IC_50_, the concentration inhibiting 50% of the cell viability; SI: selectivity index; ne: negligible effect; -: not determined.

**Table 4 molecules-27-02322-t004:** Phytochemical components of active fractions from the ethyl acetate extract of *C. speciosa* roots by LC-ESI-MS/MS.

No.	RT (min)	Detected Fraction	Tentative Identity	Precursor (*m/z*)	Exact Mass	Molecular Formula	Classification
1	4.34	SE1	4,6-Dioxoheptanoic acid	159.1	158.1	C_7_H_10_O_4_	Keto acids
2	4.41	SE3	5-(1-Hydroxyethyl)oxolan-2-one	113.1	130.1	C_6_H_10_O_3_	Lactones
3	4.65	SE4	Tri(propylene glycol)methyl ether	207.2	206.2	C_10_H_22_O_4_	Ethers
4	4.69	SE3	5-(4-Hydroxypentyl)benzene-1,3-diol	197.1	196.1	C_11_H_16_O_3_	Resorcinols
5	4.78	SE2	Indole-7-carboxaldehyde	146.1	145.1	C_9_H_7_NO	Indoles
6	5.16	SE4	Calycosin	285.1	284.1	C_16_H_12_O_5_	Isoflavones
7	5.22	SE2	Syringaresinol	401.2	418.2	C_22_H_26_O_8_	Lignans
8	5.24	SE3, SE4	Odoratin	315.1	314.1	C_17_H_14_O_6_	Isoflavones
9	5.33	SE2	Pinoresinol	341.1	358.1	C_20_H_22_O_6_	Lignans
10	5.43	SE3	Barpisoflavone A	301.1	300.1	C_16_H_12_O_6_	Isoflavones
11	5.64	SE4	Sebacic acid	185.1	202.1	C_10_H_18_O_4_	Esters
12	5.78	SE4	Liquiritigenin	257.1	256.1	C_15_H_12_O_4_	Flavanones
13	5.80	SE2, SE3	Pseudo-baptigenin	283.1	282.1	C_16_H_10_O_5_	Isoflavones
14	5.87	SE3	Formononetin	269.1	268.1	C_16_H_12_O_4_	Isoflavones
15	5.97	SE1, SE2	7-hydroxy-6,4′-dimethoxyisoflavone	299.1	298.1	C_17_H_14_O_5_	Isoflavones
16	6.14	SE1, SE2	Maackiain	285.1	284.1	C_16_H_12_O_5_	Flavones
17	6.25	SE1, SE2	Medicarpin	271.1	270.1	C_16_H_14_O_4_	Pterocarpans
18	6.51	SE2, SE4	2S-Amino-4E-pentadecene-1,3R-diol	258.3	257.3	C_15_H_31_NO_2_	Amines
19	6.60	SE1	Dimethyl azelate	217.1	216.1	C_11_H_20_O_4_	Fatty acid esters
20	7.48	SE4	9Z,11E,13E-Octadecatrienoic acid	279.2	278.2	C_18_H_30_O_2_	Fatty acyls
21	7.64	SE1, SE2, SE3	13-Keto-9Z,11E-octadecadienoic acid	295.2	294.2	C_18_H_30_O_3_	Fatty acyls
22	7.96	SE4	Friedelin	427.4	426.4	C_30_H_50_O	Triterpenoids
23	8.18	SE4	Linoleoyl ethanolamide	324.3	323.3	C_20_H_37_NO_2_	Carboximidic acids
24	8.64	SE3, SE4	Bacosine	439.4	456.4	C_30_H_48_O_3_	Triterpenes
25	8.84	SE3, SE4	Betulin	425.4	442.4	C_30_H_50_O_2_	Triterpenes
26	8.89	SE4	Ergosterol peroxide	429.4	428.3	C_28_H_44_O_3_	Ergostane steroids

RT—retention time.

## Data Availability

Data are contained within this article.

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
