# Peer review of "Cytotoxicity of *Callerya speciosa* Fractions against Myeloma and Lymphoma Cell Lines"

_molecules, 2022, doi:10.3390/molecules27072322_

Round 1
Reviewer 1 Report
This manuscript is very well-written. The methods and results are clearly presented. The authors report for the first time the cytotoxic effect of Callerya speciosa and its potential as source of new chemical structures for development of new antineoplastic drugs. The posible candidates identified are also discussed in this article. I have few comments:
- Revise title: The species name is always lowercase. Additionally I recommend adding the common name.
- It is misleading to describe in vitro toxicity upon myeloma and lymphona cell lines as equivalent to antitumor activity or antitumor properties. This shoud be revised since it objectively describe the cytotoxic activity.
- I recommend including the doxorubin results in the Table to facilitate the discussions carried out by the authors. The authors in the discussion section emphasized the toxicity of doxorubicin. In light of the findings, I invite the authors to calculate the selectivity index of the fractions and doxorubicin to strengthen their arguments.
Reviewer 2 Report
Concerning the manuscript entitled "Cytotoxicity of Callerya speciosa fractions against myeloma and lymphoma cell lines", I have with some comments, suggestions, and questions in order to improve the quality. As general comments, since it describes an un-investigated species, possessing significant cytotoxic effects against the myeloma and lymphoma cell lines, which can be of interests by other researchers and readers:
- Please revise "Callerya Speciosa" to "Callerya speciosa" in title, all the similar cases should be corrected
Abstract
L19,20: Please mention the cytotoxic activities have been analysed in this study against the selected cells with these results ....
Introduction
lack of consistency, please re-arrange this section, the statements should be arranged; the current version, followings are mentioned: the role of traditional herbal medicine, then a sentence as the aim of study (Therefore, in this study, we collected the roots of a rare medicinal plant, Callerya speciosa, to extract bioactive compounds and evaluate their pharmacological potential), the plant characteristics, cancer importance, again aim of the study; I would prefer to first speak about cancer and the selected ones in this study, then the importance of traditional medicine to cure diseases (e.g. cancer, etc.), the plant characteristics mentioning its traditional applications, biological activities, phytochemicals, etc., then the aim
- no phytochemical studies on the other species have been carried out? please describe
Results
section 2.2.: no isolation has been performed in this study, only the extraction, solvent-solvent partitioning, and fractionation have been done, please revise throughout the text
Figure 2: "ml" to "mL" please, check all the text please
L178: revise please "should not exceed" grammatically
Discussion
L279-281: I do not agree with "Consequently, these products may be more useful than synthetic medicines in the development of anticancer treatment with less toxicity related to oxidative stress", since the crude extracts/fractions contain many compounds which can cause other disorders/diseases, therefore the bioactive compounds in pure form can be as options, of course after passing many steps (in vitro, in vivo, clinical, toxicity, etc.)
- According to results (L105): "The highest TPC (1.77 mg GAE/g DW) was found in the SM extract" but "In the DPPH assay, the strongest extract was SE with a IC50 value of 0.20 mg/mL", please discuss what can be the rationales, in the discussion section
- L296,297: authors: "Therefore, isoflavones may play a major role in the anti-radical abilities of these fractions.", please bring some evidence
- please describe the main phytochemicals identified from the other species of the genus, that can be also helpful to propose the major phytochemicals of the studies species
Methods
section 4.4. since the DPPH and ABTS assays are well-known methods, I suggest skipping describing in details
Conclusion
in my opinion isolation of the major phytochemicals responsible for the tested bioactivities can be the first promising approach for further investigation, please describe
Round 2
Reviewer 2 Report
The revised version can be subjected for further publication procedure.